# Optimizing health and nutrition status of migrant construction workers consuming multiple micronutrient fortified rice in Singapore

**Femke Hannes[1], Kevin Moon[2¤], Regina Moench Pfanner****[1]\***

**1** Ibn360, Singapore, Singapore, **2** 45RICE, Singapore, Singapore

¤ Current address: Londsdale Capital, Singapore, Singapore

\* regina@ibn-360.com

## Abstract

### Introduction

A well-nourished workforce is instrumental in eradicating hunger, alleviating poverty, and spurring economic growth. A fifth of the total workforce in high-income countries are migrant workers. Despite the accessibility of nutritious foods in high-income countries, migrant workers often rely on nutrient-poor diets largely consisting of empty calories, which in turn leads to vitamin and mineral deficiency, also called hidden hunger, and resultant productivity loss. Here, we study the magnitude of hidden hunger in male migrant construction workers in Singapore and investigate the impact of consuming fortified rice for 6 consecutive months on the nutrition and health status of these workers.

### Methods

140 male migrant workers aged 20–51 years of either Bangladeshi or Indian ethnicity from a single dormitory in Singapore volunteered to participate in the study. In total, 133 blood samples were taken at the start of the study and were used to assess vitamin B12, hemoglobin, ferritin, folate, and zinc levels; a sub-sample underwent for homocysteine testing. Anthropometric measurements and vital signs, such as blood pressure, were recorded before and after the intervention.

### Results

The results show that vitamin and mineral deficiency was present, especially folate (59% of workers deficient) and vitamin B12 (7% deficient, 31% marginally deficient). The consumption of fortified rice significantly improved the vitamin, iron and zinc level in the workers and significantly reduced the systolic blood pressure amongst the Bangladeshi migrant workers, specifically.

**Data Availability Statement:** Data is available from Dryad: https://doi.org/10.5061/dryad.3n5tb2rkx

**Funding:** Please note that DSM stands for Dutch State Mines. However, DSM never writes out this

acronym and is only known under DSM. In the case of our study, it is DNP (DSM Nutrition Products) that funded the study. DSM Nutritional Products (DNP) is a manufacturer and supplier of vitamins. DNP funded the study and had provided scientific inputs to the study design and implementation, but was not involved in data analysis, interpretation and the decision to submit this manuscript. Funding for the development of this manuscript was provided by ibn360 a consultancy agency that helps the public and private sectors to co-create solutions that sustainably improve nutrition; page charges were borne by DSM DNP. The funder provided support in the form of a paid service agreement with ObvioHealth to conduct the clinical trial. RMP works at ibn360 and received financial support by the funder for consultancy. KM was affiliated at 45RICE at the time of the study, a social for-profit business by selling a product or service to the public and acquiring funds through sales. KM has no funding support to disclose. FH was employed by DNP at the time of the study. The specific roles of these authors are articulated in the 'author contributions' section.

**Competing interests:** RMP was paid for consultancy by DNP; FH was employed by DNP at the time of the study; KM has no funding support to disclose. There are no other competing interests to declare. This does not alter our adherence to PLOS ONE policies on sharing data and materials. DNP financial support had no influence on the statistical analysis which was conducted independently; and on writing of the article. The authors alone are responsible for the views expressed in this publication.

## Conclusion

Our study demonstrates that fortified rice may have a positive impact on male migrant construction worker health and nutrition status at the workplace.

## Introduction

Today, one person out of nine in the world goes hungry [1] and the number has been slowly rising since 2014 [2]. Micronutrient malnutrition, also known as hidden hunger, affects more than two billion people globally [3]. Not meeting the required nutrient intakes over a prolonged period leads to serious health consequences. In 2017, a large systematic analysis identified diet as the main risk factor for morbidity and mortality related to non-communicable diseases, overtaking tobacco use. Notably, more deaths were caused by nutrient-poor diets with low intake of foods such as whole grains, fruit, nuts, and seeds than by diets with high intake of trans fats, sugary drinks, and red and processed meats [4].

Affordability and availability of a healthy diversified diet is a major problem around the world, with a substantial part of the population consuming an energy-dense, nutrient-poor diet high in refined carbohydrates and starches and low in animal-source foods including meat, eggs and fish, or fresh vegetables and fruits. In many Asian countries, refined carbohydrates, such as white rice, form 70–80% of daily calories consumed [5]. The State of Food Security and Nutrition in the World 2020 reported that 1.5 billion people could not afford a diet that meets the required levels of essential nutrients [2]. The World Food Programme concurs that economic access is a barrier to providing a nutritious diet, finding a high percentage of households in many low- and middle-income countries unable to achieve the minimum dietary diversity required to meet nutrient needs [6]. For example, in Bangladesh, only 13% of households can afford a diet that meets their nutrient needs, and such a diet now costs twice as much as one that simply meets energy needs [6].

In South Asia in particular, malnutrition is widespread and poses a substantial threat to national economies and public health [7–9]; iron deficiency alone accounts for a loss of US$5 billion in productivity [10]. Consequently, strategies have been put in place by some governments to provide safety nets such as free meal programs to vulnerable groups. However, migrant workers who leave their country to provide a better livelihood for their family back home often come from less fortunate, food and nutrition insecure households, yet have no access to these subsidies in their adopted country. Surviving on low wages, and housed in dormitory-style accommodation with limited access to cooking facilities and refrigeration, migrant workers often rely on their employer for the provision of food [11]. Consumption of nutrient-dense food, which comes at a significant cost to the worker, may be out of reach, widening the nutrient intake gap even further.

Migrant construction work and other manual labor is integral to the economies of many high- and upper middle-income countries, where migrant workers are predominantly male, make up approximately 5% of the total world population and constitute almost 20% of the national workforce [12]. A well-nourished workforce is instrumental in the eradication of hunger and alleviation of poverty; two key United Nations Sustainable Development Goals [13]. As early as the 1960s, leading health organizations have stressed the importance of workplace nutrition to workers, employers, and governments [14–16]. According to the International Labour Office, poor diet at the workplace is costing countries around the world up to 20% in

lost productivity [15]. As much as a 30% impairment in physical work capacity and performance is reported in iron-deficient men and women [17].

Investing in human capital through improved nutrition is one way to influence both national economies and public health [13]. Workforce interventions in low- and middle-income countries involving micronutrient fortification of food have effectively addressed micronutrient deficiency in women, children and other vulnerable groups [18–24]. However, most of these interventions have been applied to female workers at risk for iron deficiency anemia or other forms of anemia, and have investigated fortification with a single micronutrient, often iron or iodine, but also vitamin A or folic acid [19, 21, 22, 24]. The World Health Organization (WHO) has published a list of micronutrients critical to public health [25]. Besides iron and zinc, vitamin A, B-complex vitamins including folate are of great concern and are well known to have a profound impact on human health and productivity, in particular the relationship of vitamin B interventions and homocysteine levels and blood pressure. Delivering micronutrients via a holistic approach and at scale at the workplace has the potential to reduce micronutrient deficiency, improve mental and physical performance of workers and consequently provides a win-win proposition for the employer, public health, and national economy.

Fortified staple food is a proven and effective way to bridge the nutrient gap at scale and ensures adequate intakes of micronutrients without the need for behavioral change [26, 27]. Rice fortified with vitamins and minerals can serve as a nutritious alternative to the highly polished, nutrient-poor white rice [27]. Since white rice is the leading staple food for Asian migrant construction workers, fortifying white rice offers a cost-effective solution and is easily prepared and implemented at the workplace.

Here, we aim to investigate the magnitude of hidden hunger in male migrant construction workers residing in Singapore and assess the impact of prolonged consumption of multiple-micronutrient-fortified rice on nutrition and health status at the workplace.

## Methods

### Study site and subjects

The study was conducted between March and September 2018 in a single dormitory in Singapore which housed approximately 180 Indian and Bangladeshi construction workers. The site is licensed in accordance with the Foreign Employee Dormitories Act which governs the working and living conditions of migrant construction workers in Singapore. The workers spend on average 6 days per week from 6 am till 6 pm on the construction site including a 1-hour break for lunch [28]. For most workers, the dormitory/site manager organizes three meals per day, 7 days per week. All meals are prepared by a designated and licensed caterer. Breakfast and lunch are prepared and delivered to the workers in the morning whereas dinner is delivered to the dormitory in the evening. Typically, each lunch and dinner consist of about 500g of cooked white rice with some vegetables and proteins [11]. A rotation of the menus is done every few weeks offering 4 different menus including vegetarian options (see S1 and S2 Tables).

Instead of the planned sample of 180 construction workers for this study, 140 construction workers consented to participate. This reduction is explained by absence of interest, motivation and personal time table conflicts. After a clinical examination and a health interview, 7 workers were excluded from the study due to pre-existing medical conditions (e.g. hepatomegaly, hyperthyroidism, under- or overweight) or chronic disease (e.g. diabetes type 2 or clinical hypertension). In total, 133 construction workers were provided with meals from the same licensed caterer, but with 500 g cooked fortified rice instead of their

highly polished white rice for 14 meals per week (lunch and dinner, 7 d per wk) over 26 consecutive weeks (approximately 6 months). All the other food items and quantities on the menu remained the same. The provision of 500 g of cooked rice per meal remained the same throughout the study and is a slight reduction from the initial estimated amount in the study protocol. The workers who participated throughout the whole study received a monthly reduction on their payment to the food caterer, equivalent to approximately 20% of their salary. The workers who did not consent to the study, did not receive the fortified rice and organized their meals by their own choice.

The TREND statement checklist was established to allow transparent reporting of the study (see S1 Checklist).

## Study design, sampling and monitoring

The design of the study is a single-center, open label, longitudinal intervention study with a before and after evaluation. The present study was conducted according to the guidelines laid down in the Declaration of Helsinki, and all procedures and amendments thereof involving human subjects were approved by the Parkway Independent Ethics Committee (PIEC/2017/019) and Agri-Food and Veterinary Authority of Singapore. The present trial was registered in the ClinicalTrials.gov Trial Registry as NCT04343508. The trial was registered after subject recruitment began due to a professional oversight; the authors confirm that all ongoing and related trials for this drug/intervention are registered. Informed written consent was obtained after an explanation of the study objectives and procedures in either Tamil or Bengali (see S2 File).

The objectives of this study were 1) to assess the extent of pre-existing nutrient deficiencies and to investigate the effectiveness of a 6-month dietary intervention with fortified rice on hemoglobin (Hb), ferritin, zinc, folate and vitamin B12 levels of migrant construction workers in Singapore and 2) examine the impact of the 6-month dietary intervention on determinants of cardiovascular disease, such as blood pressure, and for a small subset of randomly selected workers, homocysteine concentrations in blood.

At the start of the intervention study, an interview using a general health questionnaire (see health questionnaires S4a in English, S4b in Tamil and S4c in Bengali in S1 File) a clinical examination, and blood pressure and anthropometric measurements of each enrolled study subject were done onsite. We were not able to validate the health questionnaire as the migrant workers were not available due to their working schedule for any additional time except for the health assessment at the start of the intervention. Height and weight were recorded via a column scale with an adjustable measuring rod (BW2150H, Nagata Scale, Tainan City, Taiwan) followed by a Body Mass Index (BMI) calculation in line with the guidelines described by WHO [29]. Blood pressure was obtained under the supervision of a certified nurse using an electronic blood pressure monitor and measured at least three times per subject. A clinical examination and a general health interview were conducted under the supervision of a medical doctor and translator. The health interview collected data on medical history and the use of drugs and dietary supplements. The questionnaire also included questions about habits such as smoking. A blood sample was drawn onsite by a certified phlebotomist. The subject criteria for the collection of blood were defined as: 1) being between 20 and 51 years old, 2) having a BMI between 17.0 to 27.5 kg/m$^2$, 3) understanding and signing informed consent, 4) being free of concomitant medical issues and chronic disease(s), 5) having no prior usage of drugs and/or dietary supplements, and 6) smoking less than 10 cigarettes per day for a prolonged period. In addition, a minimum commitment of 6 consecutive months to the study procedure was required. Note that an amendment had been made to the original study protocol to extend the eligible age for the study from 21–50 y to 20–51 y.

During the intervention, a record was kept by the site manager including the number of pre-packed meals returned to the licensed caterer and days absent due to illness. Compliance monitoring was done at regular intervals by the project manager. A subject was considered fully compliant when having consumed all 14 meals per week offered, moderately compliant when having consumed 8–13 meals per week, or non-compliant when having consumed less than 8 meals per week. Workers who did take their home leave during the intervention would be flagged as moderate or non-compliant depending on the average consumption of meals over the whole intervention.

After 6 consecutive months of follow-up, anthropometric measurements, blood pressure readings and a blood sample were taken from the eligible construction workers.

Safety was assessed and reported according to ICH-GCP guidelines. All Adverse events (AEs) occurring during the study were reported and recorded in a Case Report Form by the study team, whether or not they were considered to be non-serious, serious, or related to the product. Definitions of safety parameters were defined in the study protocol. The AEs were classified by preferred term and body system using MedDRA and tabulated by severity and relationship. Concomitant and pre-study medications were listed by subject ID, summarized by medication class and treatment group, and listed by intervention group.

## Fortified rice (interventional product)

The fortified rice kernels produced for this study were generated using hot extrusion technology and are made from white rice flour mixed with premix. The fortified kernels are then blended with polished white rice kernels at a ratio of 1% (1:99) to obtain fortified rice ready for consumption at the required concentrations of micronutrients. Blending was done by a third-party blending company that is licensed under the Singapore laws on food production per good manufacturing practice standards. The fortified rice blend was prepared following an identical cooking method to the unfortified white rice. Each of the pre-packed meals provided to study subjects contained 500g cooked fortified rice.

The micronutrient levels per 100g uncooked fortified rice were: 150 μg vitamin A (vitamin A palmitate), 0.5 mg vitamin B1 (thiamin mononitrate), 7 mg vitamin B3 (niacin amide), 0.6 mg vitamin B6 (pyridoxine hydrochloride), 1 μg vitamin B12 (cyanocobalamin), 130 μg folic acid, 4 mg iron (ferric pyrophosphate), and 6 mg zinc (zinc oxide). Consuming their daily meals, the workers receive the following nutrient support: 500 μg vitamin A; 1.6 mg B1; 23.4 mg B3; 20 mg B6; 3.34 μg B12; 434 μg Folic Acid; 13.36 mg Fe; 20 mg Zn. This is based on 500 g cooked rice (per meal) to be approximately 167 g of uncooked rice (per meal).

## Biochemical analysis of blood samples

Eight mL of whole blood were collected from fasting subjects at baseline and after 6 months of intervention by venipuncture into an EDTA-vacutainer and a plain tube without anticoagulants (Becton Dickinson) by a certified phlebotomist. From a randomly selected subset of 53 workers, an extra 5 mL of whole blood were collected in an additional plain tube (Becton Dickinson) for determination of serum homocysteine concentration. These tubes were stored on ice (but not frozen) for a maximum of 8 h prior to arrival in the laboratory. After arrival in the laboratory, whole blood samples underwent a complete blood count and were thereafter immediately centrifuged for 10 minutes at 1000-2000g and the serum or plasma removed and stored at 2 to 8˚C until further analysis.

Hemoglobin (Hb) was measured within 24 h of sampling using an automated haematology analyzer (CELL-DYN Ruby System, Abbott Diagnostics, IL, USA) with two independent measuring channels. Serum folate, serum ferritin, vitamin B12 and homocysteine assays were run

on the Architect-2000 system (Abbott Diagnostics, IL, USA). Serum zinc was measured using inductively coupled plasma mass spectrometry on the Agilent technologies 7700 series ICP-MS instrument (Agilent Technologies, Inc., CA, USA) and trace element free tubes. Serum zinc was expressed in μg/L.

All laboratory measurements were done according to international standards.

## Definition of clinical and biochemical outcomes

Normal reference values in fasting adult males were defined as follows: serum ferritin, ≥15 μg/L [30]; vitamin B12, >220 pmol/L [31]; serum folate, ≥10 nmol/L [32]; hemoglobin (Hb), ≥13 g/dL [17]; serum zinc, ≥11.3 μmol/L [33]; and homocysteine, <15.0 μmol/L [34]. Marginal vitamin B12 level was defined as a concentration of 150–220 pmol/L [31]. Vitamin B12 deficiency was defined as a concentration <150 pmol/L.

BMI was categorized into underweight (BMI<18.5 kg/m$^2$), normal (BMI 18.5–24.9 kg/m$^2$), and overweight (BMI>25.0 kg/m$^2$) according to standard recommendations [29]. The average of all systolic blood pressure measurements taken at a single follow-up visit was used as the mean systolic blood pressure for each study subject. Similarly, the average of all diastolic blood pressure measurements was used as the mean diastolic blood pressure. Hypertension was defined as having a mean systolic blood pressure ≥140 mmHg or a mean diastolic blood pressure ≥ 90 mmHg. "At risk of hypertension" or prehypertension was defined as having a systolic blood pressure of 120–139 mmHg or a diastolic blood pressure of 80–89 mmHg [35]. Normal blood pressure was defined as having a systolic blood pressure <120 mmHg and a diastolic blood pressure of < 80 mmHg.

## Statistical analysis

A sample size of 180 participants with a maximum anticipated subject attrition of 22% was estimated based on the following: a paired t-test, a power of 80% to achieve significance on all co-primary endpoints (vitamin B12, hemoglobin, ferritin, folate, and zinc levels) simultaneously (95% power for each primary endpoint individually) and a two-sided alpha of 5% (overall), and an effect size of ≥0.30 for all parameters. The study suffered from high dropout rates; first, with 40 of the original 180 workers who did not report for screening, and second, an additional 40 of the 140 remaining participants who either left the study or declined involvement in the final study visit. The aimed sample size of 140 for an overall power of 80% was not achieved, and therefore, the study might be underpowered to detect significant effects.

For all analyses, statistical significance was defined as a P-value <0.05. SPSS version 26 software (IBM Corp., Released 2019, Armonk, NY, USA) was used for all analyses. Contrary to the study protocol only an Intent-to-treat analysis was considered for the data-analysis. Analysis of outcomes was done with the original continuous measurement values. Paired t-tests were used to compare baseline to post-intervention measurements. A two-sample or independent t-test was used to determine if mean values of subgroups were significantly different. The data have been assessed for normality of the distribution of all the change in indicators; all the indicators were sufficiently close to normally distributed to allow the use of the paired t-test.

In addition, these continuous variables were categorized according to the proportion of workers below or above the consensus cutoffs to define the presence and severity of various health conditions, including hypertension, hyperhomocysteinemia and micronutrient deficiencies. These data were not part of the planned statistical analysis and are descriptively reported in the results. The frequencies of Adverse Events (AEs) are also descriptively reported in the results.

## Results

A total of 140 subjects were screened at the start of the study; 133 subjects were enrolled and 7 subjects were excluded for not fulfilling the inclusion criteria (Fig 1). All eligible subjects were male migrants from either Bangladesh or India. About 68% of the workers were between 25–35 y old, 82% had a BMI within normal range (7.5% were underweight and 10.5% were overweight/obese), and 71% had a blood pressure below 140/90 mmHg. Vitamin deficiencies were commonly found; almost 59% of subjects were deficient in folate, and 7% were deficient and 31% marginally deficient in vitamin B12. Hyperhomocysteinemia was detected in 62% of 53 randomly selected workers (Table 1).

After 6 months of intervention, an attrition rate of 24.8% was recorded, leaving in total 100 subjects for post-intervention assessment; of these, 64 were of Bangladeshi and 36 were of Indian origin (Fig 1). In total, 14 of the 100 study subjects who had both baseline and endpoint data took home leave during the follow-up period, therefore did not consume the full complement of fortified rice. Five of these study subjects took home leave for more than 6 weeks, four study subjects took 5 to 6 weeks of home leave, and another five study subjects less than 4 weeks (S3A and S3B Tables in S3 Table). A total of 67 subjects experienced adverse events (AEs) during the study of which only 6 were moderate AEs and the remaining were considered mild. None of the AEs led to discontinuations and all AEs were resolved except for one that was ongoing at the time of study closure. AEs were generally flu, headache and similar events that were determined to be mainly unrelated to the administered product. None of the subjects experienced an AE attributed to fortified rice, and none of the subjects had a severe adverse effect (SAE).

When considering the entire sample, the vital signs of the workers did not change significantly from baseline to post-intervention (Table 1). On the other hand, there was increase in hemoglobin, ferritin, zinc, and vitamin B12. Serum folate level did not change significantly (Table 2).

Among Bangladeshi subjects, there was a significant drop in systolic blood pressure (Fig 2). In addition, there was a significant increase in the concentrations of hemoglobin, ferritin, zinc, and vitamin B12 (Table 2). Although the folate concentration increased over the course of the study, this change was not statistically significant.

Among Indian study subjects, only serum ferritin concentration demonstrated a significant increase. The average concentrations of serum ferritin and vitamin B12 at baseline were lower in Indian study subjects than Bangladeshi study subjects (Table 2). In all study subjects combined, homocysteine concentration declined with statistical significance.

The number and proportion of volunteers before and after the intervention with (pre) hypertension, hyperhomocysteinemia, anemia, and deficient for micronutrients (below or above the defined cut-off values) is summarized in Table 3. Overall, the proportion of nutrient-deficient and hypertensive workers post-intervention did not seem different as compared to baseline although this was not statistically tested. The proportion of workers with hyperhomocysteinemia seemed to be lower at post-intervention as compared to baseline but numbers were small.

## Discussion

Micronutrient deficiency was diagnosed amongst the male migrant workers at the start of our study. Six months after the consumption of fortified rice, micronutrient levels except folate levels, in all subjects, and homocysteine level in those tested, significantly improved. Systolic blood pressure significantly reduced in the Bangladeshi subjects. To our knowledge, this is the first study involving male migrant construction workers to report on a potential strategy at the

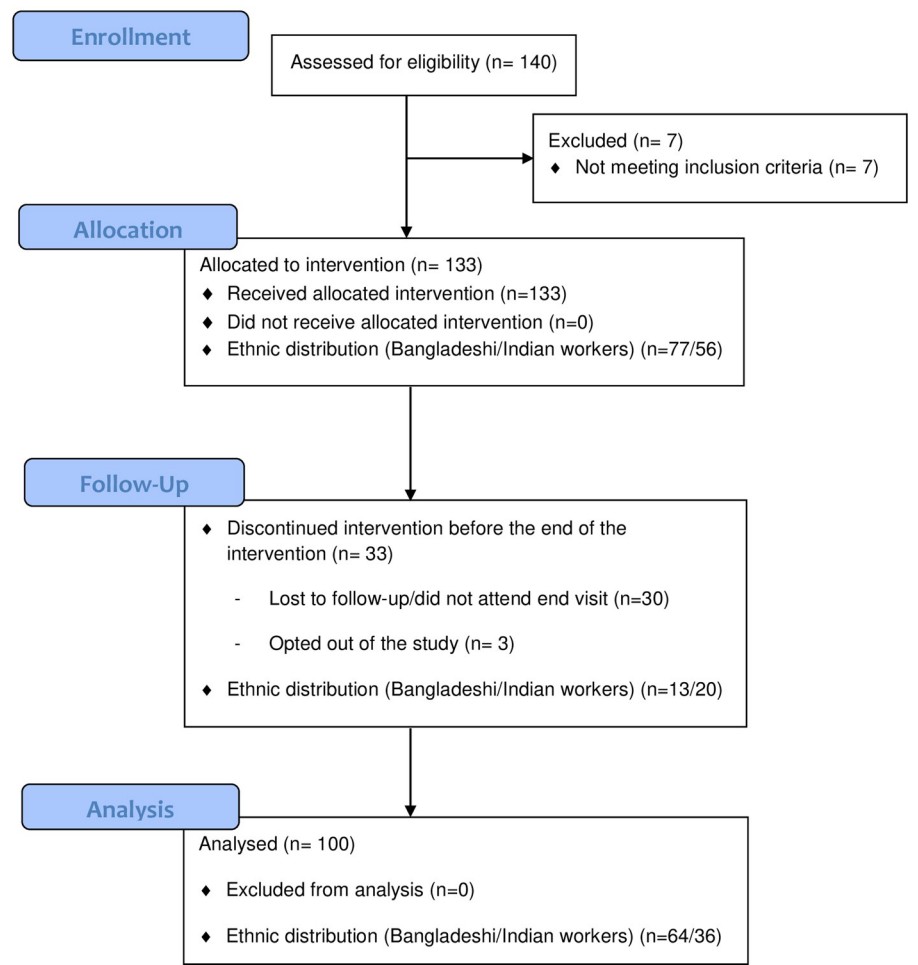

**Fig 1. Subject flow chart.**

workplace to fill the micronutrient gap, and subsequent decline in systolic blood pressure and homocysteine level, which is a risk factor for cardiovascular disease. Our study findings fill an important gap in the literature pertaining to male migrant construction workers health and nutrition status at the workplace.

## Tackling the nutrient gap with fortified rice at the workplace

Folate and vitamin B12 deficiencies were commonly diagnosed in the subjects at the start of the study. Moreover, folate deficiency was more prevalent amongst the Bangladeshi workers whereas vitamin B12 deficiency was more often diagnosed amongst the Indian workers.

**Table 1. Overview of the nutrition, blood pressure, and homocysteine status of 133 eligible construction workers.**

| Total | All eligible subjects N = 133 | |
|---|---|---|
| | n | % |
| Hypertensive | 39 | 29% |
| At risk for hypertension | 0 | 0% |
| Normotensive | 94 | 71% |
| Anemia (Hb <13 g/dL) | 1 | 1% |
| Iron deficient (ferritin <15 μg/L) | 5 | 4% |
| Folate deficient (<10 nmol/L) | 78 | 59% |
| Vitamin B12 deficient (<150 pmol/L) | 9 | 7% |
| Vitamin B12 marginal level (150–220 pmol/L) | 41 | 31% |
| Zinc deficient (<11.3 μmol/L) | 7 | 5% |
| | N = 53* | |
| Homocysteinemia (homocysteine ≥15.0 μmol/L) | 33 | 62% |

*A subgroup of 53 construction workers were tested for homocysteine level at baseline which is a slight modification to the original study protocol that estimated to draw blood from 50 construction workers

Although their meals typically contained a few tablespoons of stewed or fried vegetables such as lentils, cauliflower, peas, brinjal, potato, bitter gourd or cabbage, some of which are rich sources of folate, prolonged cooking is known to affect folate retention of these foods [36]. Of the Indian workers, 44% consumed a strict vegetarian diet, whereas none of the Bangladeshi workers did. Presumably, their food and nutrition insecure backgrounds, the prolonged and restricted offering of low-quality vegetables or meat and the increased need for B-vitamins and other micronutrients to sustain physically demanding jobs are the root causes for these poor outcomes.

Multi-micronutrient-fortified rice was used to fill the micronutrient gap at the workplace by replacing the nutrient-poor white rice with nutrient-dense fortified rice. Each subject was provided with 2 meals per day containing fortified rice, providing the workers between 1 to 2 times the recommended nutrient intake for all vitamins and minerals included [37].

Despite the increment in daily nutrient consumption, the mean folate and vitamin B12 level remained relatively unchanged in all subjects combined and the Indian subjects, respectively. These findings suggest that 1) poor absorption of vitamin B12 and folate could cause this nutrition deficit to persist in these workers, 2) there might be a need for increased requirements due to a pre-existing nutritional deficit upon arrival at the workplace, 3) there might be a need for increased requirements due to intensity of physical labor or 4) the workers may carry a genetic predisposition for common polymorphisms in gene coding for proteins involved in the one-carbon metabolism. Clinical evidence reports that polymorphisms in methylenetetrahydrofolate reductase and methionine synthase exist in Indians and correlate with decreased serum folate, vitamin B12 and elevated homocysteine level [38–42]. Moreover, McNulty and colleagues mention a potential benefit of folate and/or methylenetetrahydrofolate supplementation with riboflavin (a co-factor of the methylenetetrahydrofolate reductase enzyme) in their folate biomarker response which in turn can lower blood pressure, particularly in adults with an impaired one-carbon metabolism [43]. Further studies are required to elucidate the specific benefits of the intervention with methylenetetrahydrofolate instead of—or in combination with—folic acid and including riboflavin in establishing a healthier blood pressure.

**Table 2. Evaluation of the impact of the fortified rice intervention on the vital signs and biochemical nutritional status for all 100 subjects completing the study, and Bangladeshi and Indian subgroups.**

| | All subjects n = 100 | | | Bangladeshi subjects n = 64 | | | Indian subjects n = 36 | | |
|---|---|---|---|---|---|---|---|---|---|
| | Mean (SD) | Mean change | P-value* | Mean (SD) | Mean change | P-value | Mean (SD) | Mean change | P-value |
| **Pulse (beats/min)** | 70.4 (10.4) | -0.27 | 0.79 | 71.4 (11.0) | -2.01 | 0.123 | 68.5 (9.1) | 2.84 | 0.05 |
| **Temperature (˚C)** | 36.3 (0.3) | 0.04 | 0.36 | 36.3 (0.2) | -0.07 | 0.061 | 36.1 (0.35) | 0.22 | 0.01 |
| **Systolic blood pressure (mmHg)** | 126.6 (13.4) | -1.56 | 0.18 | 125.6 (12.6) | -3.81 | 0.005 | 128.5 (14.8) | 2.44 | 0.25 |
| **Diastolic blood pressure (mmHg)** | 82.3 (10.9) | 0.45 | 0.63 | 80.9 (10.2) | -0.66 | 0.475 | 84.9 (11.8) | 2.44 | 0.22 |
| **Hb (g/dL)** | 15.5 (1.1) | 0.28 | <0.001 | 15.1 (0.9) | 0.33 | <0.001 | 15.8 (1.4) | 0.19 | 0.07 |
| **Folate (nmol/L)** | 9.2 (3.9) | 0.07 | 0.79 | 8.5 (4.2) | 0.54 | 0.11 | 10.55 (3.1) | -0.76 | 0.07 |
| **Ferritin (μg/L)** | 107.5 (78.1) | 10.77 | <0.001 | 122 (79.6) | 9.19 | 0.03 | 80.1 (68.1) | 13.58 | 0.01 |
| **Vitamin B12 (pmol/L)** | 280.7 (103.5) | 20.49 | <0.001 | 307.3 (95.3) | 34.02 | <0.001 | 233.6 (101.8) | -3.56 | 0.46 |
| **Zinc (μmol/L)** | 14.3 (1.7) | 0.55 | 0.02 | 14.4 (1.62) | 0.71 | 0.02 | 14.0 (1.9) | 0.28 | 0.44 |
| **Homocysteine (μmol/L)\*\*** | 15.1 (3.5) | -2.58 | <0.001 | 14.6 (4.1) | -3.10 | <0.001 | 16.0 (2.6) | -1.77 | 0.04 |

\* P-values are calculated as one sample t-test, 2-tailed

\*\* Values calculated for only n = 18 subjects with available values; n = 11 Bangladeshi, and n = 7 Indian subjects

## The impact of improved workplace nutrition on health

Elevated homocysteine levels have been identified as a relevant biomarker of cardiovascular disease [44]. In our study, the prevalence of hyperhomocysteinemia declined significantly after the consumption of fortified rice. Literature suggests that 5–10% of the general population presents with elevated homocysteine levels in serum [45]. However, in the South Asian population, the prevalence of hyperhomocysteinemia may be as high as 30% to 60% [41, 46]. Studies have shown that when homocysteine levels are elevated by 5 μmol/L, the risk of cerebrovascular disease increases by 59%, and the risk of coronary heart disease increases by 32% [44]. The reduction of homocysteine levels by 3 μmol/L reduces the risk of stroke by 24%, and the risk of ischemic heart disease by 16% [44]. Furthermore, population data and randomized trials have provided strong evidence that an intervention with B-vitamins, such as folic acid alone or in combination with other B-vitamins can significantly reduce stroke risk [47]. In 2017, a Cochrane review reported on an effect favoring interventions with vitamins B6, B12 and folic

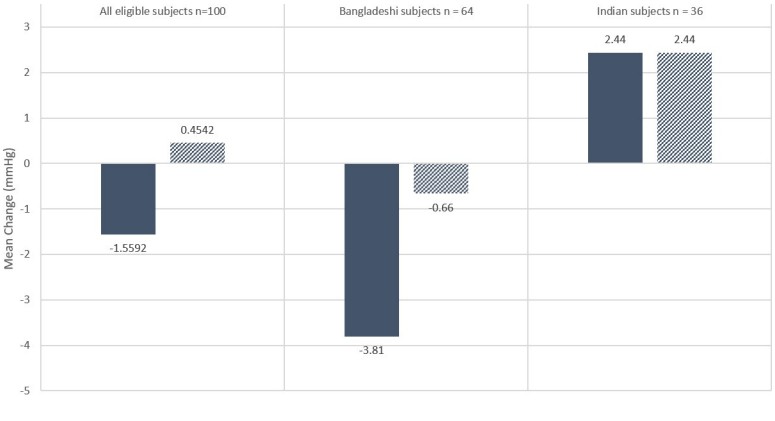

**Fig 2. Mean change in blood pressure status after the study intervention.**

**Table 3. Baseline and post-intervention characteristics of blood pressure, micronutrient deficiencies and hyperhomocysteinemia (number and percentage of participants).**

| Total | | All eligible subjects N = 100 | | Bangladeshi subjects N = 64 | | Indian subjects N = 36 | |
|---|---|---|---|---|---|---|---|
| | | n | % | n | % | n | % |
| **Hypertensive** | Baseline | 29 | 29% | 15 | 23% | 14 | 39% |
| | Endline | 23 | 23% | 10 | 16% | 13 | 36% |
| **At risk for hypertension** | Baseline | 41 | 41% | 28 | 44% | 13 | 36% |
| | Endline | 47 | 47% | 30 | 47% | 17 | 47% |
| **Normotensive** | Baseline | 30 | 30% | 21 | 33% | 9 | 25% |
| | Endline | 30 | 30% | 24 | 38% | 6 | 17% |
| **Anemia (Hb <13 g/dL)** | Baseline | 0 | 0% | 0 | 0% | 0 | 0% |
| | Endline | 0 | 0% | 0 | 0% | 0 | 0% |
| **Iron deficient (ferritin <15 µg/L)** | Baseline | 2 | 2% | 2 | 3% | 0 | 0% |
| | Endline | 1 | 1% | 1 | 2% | 0 | 0% |
| **Folate deficient (<10 nmol/L)** | Baseline | 64 | 64% | 47 | 73% | 17 | 47% |
| | Endline | 61 | 61% | 43 | 67% | 18 | 50% |
| **Vitamin B12 deficient (<150 pmol/L)** | Baseline | 7 | 7% | 1 | 2% | 6 | 17% |
| | Endline | 7 | 7% | 1 | 2% | 6 | 17% |
| **Vitamin B12 marginal level (150–220 pmol/L)** | Baseline | 33 | 33% | 12 | 19% | 21 | 58% |
| | Endline | 31 | 31% | 11 | 17% | 20 | 56% |
| **Zinc deficient (<11.3 µmol/L)** | Baseline | 4 | 4% | 1 | 2% | 3 | 8% |
| | Endline | 5 | 5% | 2 | 3% | 3 | 8% |
| | | N = 18 | | N = 11 | | N = 7 | |
| **Hyperhomocysteinemia (≥15.0 µmol/L)** | Baseline | 11 | 61% | 6 | 56% | 5 | 71% |
| | Endline | 5 | 28% | 2 | 18% | 3 | 43% |

acid which reduces homocysteine concentration and thus lower the risk of stroke [48]. These findings would suggest that through our intervention with fortified rice containing vitamin B complex, which led to an average 3 µmol/L reduction in homocysteine, a stroke risk reduction of 24% could be predicted.

In addition, a significant reduction in systolic blood pressure was reported amongst the Bangladeshi subjects. Scientific evidence shows a clear benefit of reducing blood pressure on cardiovascular health [49]. Even modest changes such as a 2 mmHg reduction in systolic blood pressure can decrease cardiovascular disease risk by 10% [50]. These findings suggest that the approximately 4 mmHg lowering of the systolic blood pressure observed after the intervention translates to an estimated 15% reduction in cardiovascular disease risk.

The prevalence of hypertension (>140/90 mmHg) amongst the migrant construction workers at start of the study was high at 29% and dropped by 6% after the intervention with fortified rice. This drop was more pronounced in the subgroup of Bangladeshi workers (-7%) compared to the Indian subgroup (-3%). This high prevalence of hypertension has been reported by other studies in young male South Asian migrant workers living abroad [51–53].

Further attention is warranted to address the nutrition deficits and health issues of male migrant construction workers in Singapore—increasing awareness amongst employers and the relevant health agencies, providing potential treatment, and working towards the prevention of future deficiencies. These findings showcase that with the introduction of nutrient-dense foods such as fortified rice at the workplace, benefits for both the workers and employers can be achieved.

## Strengths and limitations of the study

One limitation of our study is the absence of a control group which means that it was not possible to adjust for existing confounding variables. A decision was made to not include a control group from another dormitory as the variability in conditions due to differing dormitory management and employer standards would have introduced bias into our study, while individual random allocation to control and intervention group was not feasible logistically. All the subjects involved came from the one dormitory whereby subjects were well taken care of, with the same diet and lifestyle and were one coherent group. However, these subjects may not be representative of Singapore's migrant construction worker population as a whole, limiting our ability to extrapolate findings to the entire population. Another weakness of our study is the inaccurate prediction of iron status, as we did not include measurement of extra markers for inflammation. Therefore, iron deficiency may be underestimated. Another limitation of the study is that the aimed sample size of 140 for an overall power of 80% was not achieved, and therefore, the study might be underpowered to detect significant effects.

## Conclusion

This study has shown that hidden hunger is highly prevalent amongst male migrant construction workers in a single dormitory in Singapore. Improving the nutrient density of meals provided to these men at the workplace has been demonstrated to successfully improve their nutritional status and may lower their risk of cardiovascular disease through a reduction in systolic blood pressure and homocysteine level. Furthermore, our findings suggest that thorough health assessments of these workers including micronutrient status should be considered beyond the standard clinical assessments, and that complementary nutrition strategies should be implemented to fill pre-existing nutrient intake gaps. Our study demonstrates that multi-micronutrient-fortified rice may be a suitable vehicle to improve the nutrition status, to bridge these nutrient gaps and systematically eradicate hidden hunger in this population. More research is warranted to further investigate the impact of multi-micronutrient-fortified rice on the health and productivity of male migrant construction workers in Singapore on a larger scale to strengthen the evidence base for potential workplace nutrition programs.

## Supporting information

**S1 Checklist. TREND statement checklist.**
(PDF)

**S1 Table. Weekly menu catered to Bangladeshi workers.**
(PDF)

**S2 Table. Weekly menu catered to Indian workers.**
(PDF)

**S3 Table. a**. Total number of workers on home leave. * subjects do have both baseline and post-intervention. **b**. Weekly distribution of workers who took home leave. * subjects do have both baseline and post-intervention.
(ZIP)

**S1 File. a** English version of health questionnaire. **b** Tamil version of health questionnaire. **c** Bengali version of health questionnaire.
(ZIP)

**S2 File. Study protocol.**
(PDF)

## Acknowledgments

We thank the migrant construction workers and their supervisors to have participated in this study. We particularly want to thank Yannick Foing and the DSM Nutrition Products (DNP) Singapore team who were instrumental in bringing the different partners together and making this study happen. We also thank the DSM Nutrition Products (DNP) Kaiseraugst team, in particular Maaike Bruins. We are also grateful to the different health staff who were involved in the blood sample collection, laboratory analysis and overall data collection and their respective laboratories and institutions in Singapore. Specific thanks go to SPRIM and OBVIO HEALTH. We are particularly thankful to Bradly A. Woodruff who performed the external statistical analysis and reviewed several versions of the article. We also thank Fabian Rohner for his input to and review of the article.

## Author Contributions

**Conceptualization:** Femke Hannes, Kevin Moon, Regina Moench Pfanner.

**Data curation:** Femke Hannes, Regina Moench Pfanner.

**Formal analysis:** Femke Hannes, Regina Moench Pfanner.

**Funding acquisition:** Regina Moench Pfanner.

**Investigation:** Femke Hannes, Kevin Moon, Regina Moench Pfanner.

**Methodology:** Femke Hannes, Kevin Moon, Regina Moench Pfanner.

**Project administration:** Femke Hannes, Kevin Moon.

**Resources:** Femke Hannes, Regina Moench Pfanner.

**Supervision:** Femke Hannes, Kevin Moon, Regina Moench Pfanner.

**Validation:** Femke Hannes, Regina Moench Pfanner.

**Visualization:** Femke Hannes, Regina Moench Pfanner.

**Writing – original draft:** Femke Hannes, Regina Moench Pfanner.

**Writing – review & editing:** Femke Hannes, Kevin Moon, Regina Moench Pfanner.

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
