## [Decision Letter · Decision Letter 0]

18 Feb 2022

PONE-D-21-12903Optimizing health and nutrition status of migrant construction workers consuming multiple micronutrient fortified rice in SingaporePLOS ONE

Dear Dr. Moench Pfanner,

Thank you for submitting your manuscript to PLOS ONE. After careful consideration, we feel that it has merit but does not fully meet PLOS ONE’s publication criteria as it currently stands. Therefore, we invite you to submit a revised version of the manuscript that addresses the points raised during the review process.

The manuscript has been evaluated by two reviewers, and their comments are available below.

The reviewers have raised a number of major concerns. They request copyediting to ensure the language and text of the study is more readable and easy to follow. The reviewers also note concerns about the statistical analyses presented, and they request more appropriate tests to be used and a re-analyses of the data to be completed.

Could you please carefully revise the manuscript to address all comments raised?

We look forward to receiving your revised manuscript.

Kind regards,

Jamie Royle

Associate Editor

PLOS ONE

Journal Requirements:

2. Thank you for submitting your clinical trial to PLOS ONE and for providing the name of the registry and the registration number. The information in the registry entry suggests that your trial was registered after patient recruitment began. PLOS ONE strongly encourages authors to register all trials before recruiting the first participant in a study.

a) your reasons for your delay in registering this study (after enrolment of participants started);

b) confirmation that all related trials are registered by stating: “The authors confirm that all ongoing and related trials for this drug/intervention are registered".

3. Please include additional information regarding the survey or questionnaire used in the study and ensure that you have provided sufficient details that others could replicate the analyses. For instance, if you developed a questionnaire as part of this study and it is not under a copyright more restrictive than CC-BY, please include a copy, in both the original language and English, as Supporting Information. If the original language is written in non-Latin characters, for example Amharic, Chinese, or Korean, please use a file format that ensures these characters are visible.

4. Please state whether you validated the questionnaire prior to testing on study participants. Please provide details regarding the validation group within the methods section.

5. Please expand the acronym “DSM” (as indicated in your financial disclosure) so that it states the name of your funders in full.

6. Thank you for stating the following in the Competing Interests section: "The authors have declared that no competing interests exists for this publication. However, two authors have been supported financially in the beginning of the study by either being an employee of DSM (FH) or an independent consulting firm (RMP) to give input to the study design. At the time of the study implementation one author (KM) was working for 45RICE, a social enterprise dedicated to improving lives through nutrition. DSM’s financial support had no influence on the statistical analysis which was conducted independently and on writing of the article. The authors alone are responsible for the views expressed in this publication."

We note that one or more of the authors have an affiliation to the commercial funders of this research study: Ibn360 and 45RICE

7. We note that you have indicated that data from this study are available upon request. PLOS only allows data to be available upon request if there are legal or ethical restrictions on sharing data publicly. For more information on unacceptable data access restrictions, please see http://journals.plos.org/plosone/s/data-availability#loc-unacceptable-data-access-restrictions. 

Reviewers' comments:

Reviewer's Responses to Questions

**Comments to the Author**

1. Is the manuscript technically sound, and do the data support the conclusions?

Reviewer #1: No

Reviewer #2: No

2. Has the statistical analysis been performed appropriately and rigorously? 

Reviewer #1: No

Reviewer #2: No

3. Have the authors made all data underlying the findings in their manuscript fully available?

Reviewer #1: No

Reviewer #2: Yes

4. Is the manuscript presented in an intelligible fashion and written in standard English?

Reviewer #1: No

Reviewer #2: No

5. Review Comments to the Author

Reviewer #1: A clinical trial was conducted which aimed to study the impact of consuming fortified rice for six months on nutrient levels and vital health status of migrant workers in Singapore. A high percentage of migrant workers exhibited folate and vitamin B12 deficiencies. Consuming fortified rice increased vitamin, iron, and zinc levels. In the subset of Bangladeshi migrant workers, a reduction in systolic blood pressure was observed from pre to post-intervention.

Major revisions:

1- Thoroughly proofread the manuscript. In many instances, phraseology is non-standard.

2- An underlying assumption of the chi-square test is independent samples. Thus, it is inappropriate to compare baseline and post-intervention proportions (percentages) using the chi-square test.

3- Include p-values in the text to support claims of statistical significance.

Minor revisions:

1- Abstract: The term “constitute” in the second sentence is used inappropriately.

2- Clarify the use of the terms “status” and “concentrations” in the abstract. Perhaps “levels” is a more precise term.

3- Abstract: Provide percentages of those with nutrient deficiencies.

4- Line 221: Replace the following two sentences with the suggestion below. “To minimize the inter-individual component covariance, the difference between baseline and endline measurements for each individual survey subject was calculated. P values were calculated using a one-sample t-test comparing the average difference to zero.” Suggested replacement: “Paired t-tests were used to compare baseline to post-intervention measurements.” Revise line 260 accordingly.

5- Normal distribution of data is an underlying assumption of the paired or independent t-test. Indicate if the data was checked for normality prior to applying these tests.

5- The term “endpoint” is unclear. Perhaps "post-intervention" is better.

6- Line 253: Revise: When considering the entire sample, the vital signs of the workers did not change significantly from baseline to post-intervention.

7- Line 255: Revise to: increase in the hemoglobin, ferrin, zinc, and vitamin B12 levels.

8- P-values never equal zero. Express small p-values as p < 0.001.

9- Lines 263, 266, 269, 270, 271: Within the text, provide the p-values to support the claims.

10- Throughout the manuscript, provide the frequencies that correspond to the percentages.

11- State and justify the study’s target sample size with a pre-study statistical power calculation.

The power calculation should include: (1) the estimated outcomes in each group; (2) the α (type I) error level; (3) the statistical power (or the β (type II) error level); (4) the target sample size and (5) for continuous outcomes, the standard deviation of the measurements.

Reviewer #2: Abstract - Need data to be showed to indicate the improvement of nutrtional status (as mentioned)

Introduction - 1.Explain the relationship between nutrient deficiencies in relation to homocystein and blood pressure.

2. Are there the previous studies / data of nutritional status of male worker in Singapore ? why is important compare to other group of population?

Method 1. Expain about how to select the study site ? and sample size calculation ?

2. Participant was given 500 g cooked rice per day or per meal (line no .119,126)

3. Line 181-184 ...100 g uncooked rice equal to how much for cooked rice?

4. Explain about the nutrients level which participant was recieved form fortified rice / day

5. Blood collection for serum Zn need to be collected, prepared with special condition to prevent contatimination e.g. all supplied used for blood collection need to be the trace element free apparatus (https://www.izincg.org/new-blog-1/2021/1/10/comparison-of-laboratory-instrument-types-for-analysis-of-plasma-or-serum-zinc-concentration) ... For my opinion, results of serum Zn may not be reliable.

6. How to control confounding factors e.g. life stye, diet, genetic

7. What 's the QC procedure / or data of refference materials for the biochemical data.

8. Why the author used independent t-test ? (since participants are the same group)

Results

1. Data need to be revised if statistical analysis would change

2. Restults of serum Zn should be removed since they not reliable

3. I would suggest that the data should be presented for both pre - and post-in the same table / figure

and the text need to be changed accordingly

4. The text need to be re-write to be more clearly and logically manner (make it mor easy to understand)

Discussion and conclusion

1. Line 287. What's the indicator which show " micronutrient deficiency was commonly reported amongst the male migrant worker" line 374 "This study has shown that hidden hunger is highly prevalent" in this study

2. Any other underlining cause or confounding factor contributed in the results e.g. life stye, diet, genetic

6. PLOS authors have the option to publish the peer review history of their article (what does this mean?). If published, this will include your full peer review and any attached files.

Reviewer #1: No

Reviewer #2: No

---

## [Author Response · Author response to Decision Letter 0]

16 Jun 2022

We have answered to all the reviewer's comments in our rebuttal letter, titled: Response to Reviewers

---

## [Decision Letter · Decision Letter 1]

18 Jul 2022

PONE-D-21-12903R1Optimizing health and nutrition status of migrant construction workers consuming multiple micronutrient fortified rice in SingaporePLOS ONE

Dear Dr. Moench Pfanner,

Thank you for submitting your manuscript to PLOS ONE. After careful consideration, we feel that it has merit but does not fully meet PLOS ONE’s publication criteria as it currently stands. Therefore, we invite you to submit a revised version of the manuscript that addresses the points raised during the review process. The concerns noted by the reviewers have been addressed. However, during my own review of the manuscript I noted some potential ethical concerns that should be addressed before the manuscript is published: Your clinical trial was conducted in a dormitory. Your manuscript indicates that 140 of the ~180 members of the dormitory consented to participate in the study. Please update your manuscript to indicate what food individuals who did not consent to take part received. Did they also receive the fortified rice? If individuals that did not consent to take part were given the fortified rice, despite not consenting to take part in the study, please discuss how the associated ethical concerns with this were mitigated.

We look forward to receiving your revised manuscript.

Kind regards,

George Vousden

Deputy Editor in Chief

PLOS ONE

Reviewers' comments:

Reviewer's Responses to Questions

**Comments to the Author**

1. If the authors have adequately addressed your comments raised in a previous round of review and you feel that this manuscript is now acceptable for publication, you may indicate that here to bypass the “Comments to the Author” section, enter your conflict of interest statement in the “Confidential to Editor” section, and submit your "Accept" recommendation.

Reviewer #1: (No Response)

2. Is the manuscript technically sound, and do the data support the conclusions?

Reviewer #1: Yes

3. Has the statistical analysis been performed appropriately and rigorously? 

Reviewer #1: Yes

4. Have the authors made all data underlying the findings in their manuscript fully available?

Reviewer #1: Yes

5. Is the manuscript presented in an intelligible fashion and written in standard English?

Reviewer #1: Yes

6. Review Comments to the Author

Reviewer #1: (No Response)

7. PLOS authors have the option to publish the peer review history of their article (what does this mean?). If published, this will include your full peer review and any attached files.

Reviewer #1: No

---

## [Author Response · Author response to Decision Letter 1]

30 Jul 2022

We have addressed the Reviewers comment and have edited the manuscript accordingly.

---

## [Decision Letter · Decision Letter 2]

28 Sep 2022

PONE-D-21-12903R2Optimizing health and nutrition status of migrant construction workers consuming multiple micronutrient fortified rice in SingaporePLOS ONE

Dear Dr. Moench Pfanner,

Thank you for submitting your manuscript to PLOS ONE. After careful consideration, we feel that it has merit but does not fully meet PLOS ONE’s publication criteria as it currently stands. Therefore, we invite you to submit a revised version of the manuscript that addresses the points raised during the review process. Please see below for additional comments from the editor that require a response.

We look forward to receiving your revised manuscript.

Kind regards,

Callam Davidson

Editorial Office

PLOS ONE

Additional Editor Comments:

The following content currently found in your Financial Disclosure should be relocated to your Competing Interests: ‘RMP works at ibn360 and received financial support by the funder for consultancy. KM was affiliated at 45RICE at the time of the study, a social for-profit business by selling a product or service to the public and acquiring funds through sales. FH was employed by DNP at the time of the study.’

Please confirm whether the informed consent was written or verbal, and include this information in your Methods.

Line 77: Please update ‘1960’s’ to ‘1960s’.

Line 121: Please update ‘andS2’ to ‘and S2’.

Your TREND checklist is missing details of how sample size was calculated, however this information is contained within the manuscript (at lines 238-245). Please update your checklist to reflect this.

Please also cite your TREND checklist in the Methods.

Your protocol mentions adverse event monitoring, but this is not reported in the manuscript – please update your manuscript and TREND checklist to include this information.

There is a discrepancy between the protocol and clinical trail registry information at https://clinicaltrials.gov/ct2/show/NCT04343508. Your primary and secondary endpoints do not match. Please explain this discrepancy and ensure the correct information is reported in the manuscript. Primary and secondary outcomes ought to be clearly demarcated in the Abstract, Methods, and the Results.

Line 204: Please state the number of randomly selected workers from whom additional blood was planned for collection (I believe this information is located at line 270).

Line 245: The underpowered nature of the study needs to be stated as a limitation in your Discussion.

Line 248: Please include the rationale for this deviation from the protocol.

Line 298: Please updated to ‘statistically significant’.

Line 305: Please justify the decision not to perform statistical analysis. I do not feel that the data presented in Table 3 necessarily support the statements on lines 304-307.

Table 3: Please consider including gridlines to make this table easier to read.

Line 320: Please avoid overstating conclusions based on the findings presented – please temper the statement ‘an effective strategy’ by adding ‘potentially’, or similar.

Line 411: In a similar vein, please temper or remove the statement ‘multi-micronutrient-fortified rice may be the ideal vehicle to optimize the health status’, as the findings from this study cannot support this claim.

Line 426: From the information presented, it appears BAW may qualify for authorship based on ICMJE criteria. Please review https://journals.plos.org/plosone/s/authorship and determine whether BAW ought to be included as an author.

Reviewers' comments:

Reviewer's Responses to Questions

**Comments to the Author**

1. If the authors have adequately addressed your comments raised in a previous round of review and you feel that this manuscript is now acceptable for publication, you may indicate that here to bypass the “Comments to the Author” section, enter your conflict of interest statement in the “Confidential to Editor” section, and submit your "Accept" recommendation.

Reviewer #1: All comments have been addressed

2. Is the manuscript technically sound, and do the data support the conclusions?

Reviewer #1: (No Response)

3. Has the statistical analysis been performed appropriately and rigorously? 

Reviewer #1: (No Response)

4. Have the authors made all data underlying the findings in their manuscript fully available?

Reviewer #1: (No Response)

5. Is the manuscript presented in an intelligible fashion and written in standard English?

Reviewer #1: (No Response)

6. Review Comments to the Author

Reviewer #1: (No Response)

7. PLOS authors have the option to publish the peer review history of their article (what does this mean?). If published, this will include your full peer review and any attached files.

Reviewer #1: No

---

## [Author Response · Author response to Decision Letter 2]

31 Oct 2022

We have responded to the reviewer's comments in our letter Response to Reviewers 31 October

---

## [Decision Letter · Decision Letter 3]

16 Jan 2023

PONE-D-21-12903R3Optimizing health and nutrition status of migrant construction workers consuming multiple micronutrient fortified rice in SingaporePLOS ONE

Dear Dr. Moench Pfanner,

Thank you for submitting your manuscript to PLOS ONE. After careful consideration, we feel that it has merit but does not fully meet PLOS ONE’s publication criteria as it currently stands. Therefore, we invite you to submit a revised version of the manuscript that addresses the points raised during the review process.

Thank you very much for your continued work on revising the submitted manuscript. The manuscript has been evaluated by a reviewer and they have provided minor comments which may be seen below. 

Could you please revise the manuscript to carefully address the concerns raised?

We look forward to receiving your revised manuscript.

Kind regards,

Richard Ali

Associate Editor

PLOS ONE

Journal Requirements:

Reviewers' comments:

Reviewer's Responses to Questions

**Comments to the Author**

1. If the authors have adequately addressed your comments raised in a previous round of review and you feel that this manuscript is now acceptable for publication, you may indicate that here to bypass the “Comments to the Author” section, enter your conflict of interest statement in the “Confidential to Editor” section, and submit your "Accept" recommendation.

Reviewer #1: (No Response)

2. Is the manuscript technically sound, and do the data support the conclusions?

Reviewer #1: Yes

3. Has the statistical analysis been performed appropriately and rigorously? 

Reviewer #1: Yes

4. Have the authors made all data underlying the findings in their manuscript fully available?

Reviewer #1: Yes

5. Is the manuscript presented in an intelligible fashion and written in standard English?

Reviewer #1: Yes

6. Review Comments to the Author

Reviewer #1: Minor Revisions:

1- Tables 1 and 3: Remove the equals sign in the table header "n=".

2- Line 304 contains a grammatical error; the verb is missing. "and none of the subjects a severe adverse effect (SAE)."

3- Line 437: Replace "pick up" with "detect".

Note that line numbers refer to those in the track changes version of the manuscript.

7. PLOS authors have the option to publish the peer review history of their article (what does this mean?). If published, this will include your full peer review and any attached files.

Reviewer #1: No

---

## [Author Response · Author response to Decision Letter 3]

18 Jan 2023

We have responded to the reviewer's comments and edited the manuscript accordingly.

---

## [Editor Report · Decision Letter 4]

2 May 2023

Optimizing health and nutrition status of migrant construction workers consuming multiple micronutrient fortified rice in Singapore

PONE-D-21-12903R4

Dear Dr. Moench Pfanner,

We’re pleased to inform you that your manuscript has been judged scientifically suitable for publication and will be formally accepted for publication once it meets all outstanding technical requirements.

Kind regards,

James Mockridge

Staff Editor

PLOS ONE

Additional Editor Comments (optional):

Abstract, line 36: in view of a comment in a previous decision regarding overstating conclusions, please ensure that this is also attended to in the Abstract. Please change to: Our study demonstrates that fortified rice may have a positive impact on male migrant...."

Please make this change before supplying the final files for your manuscript.

---

## [Editor Report · Acceptance letter]

22 May 2023

PONE-D-21-12903R4 

Optimizing health and nutrition status of migrant construction workers consuming multiple micronutrient fortified rice in Singapore 

Dear Dr. Moench Pfanner:

I'm pleased to inform you that your manuscript has been deemed suitable for publication in PLOS ONE. Congratulations! Your manuscript is now with our production department. 

Kind regards, 

on behalf of

Dr James Mockridge 

Staff Editor

PLOS ONE